# The Role of miR-128 in Neurodegenerative Diseases

**DOI:** 10.3390/ijms24076024

**Published:** 2023-03-23

**Authors:** Marika Lanza, Salvatore Cuzzocrea, Salvatore Oddo, Emanuela Esposito, Giovanna Casili

**Affiliations:** Department of Chemical, Biological, Pharmaceutical and Environmental Sciences, University of Messina, Viale Ferdinando Stagno D’Alcontres, 31-98166 Messina, Italy

**Keywords:** neurodegenerative disease, miR-128, Alzheimer’s disease, Huntington’s disease, Parkinson’s disease, brain

## Abstract

Several neurodegenerative disorders are characterized by the accumulation of misfolded proteins and are collectively known as proteinopathies. Alzheimer’s disease (AD), Parkinson’s disease (PD), and Huntington’s disease (HD) represent some of the most common neurodegenerative disorders whose steady increase in prevalence is having a major socio-economic impact on our society. Multiple laboratories have reported hundreds of changes in gene expression in selective brain regions of AD, PD, and HD brains. While the mechanisms underlying these changes remain an active area of investigation, alterations in the expression of noncoding RNAs, which are common in AD, PD, and HD, may account for some of the changes in gene expression in proteinopathies. In this review, we discuss the role of miR-128, which is highly expressed in mammalian brains, in AD, PD, and HD. We highlight how alterations in miR-128 may account, at least in part, for the gene expression changes associated with proteinopathies. Indeed, miR-128 is involved, among other things, in the regulation of neuronal plasticity, cytoskeletal organization, and neuronal death, events linked to various proteinopathies. For example, reducing the expression of miR-128 in a mouse model of AD ameliorates cognitive deficits and reduces neuropathology. Overall, the data in the literature suggest that targeting miR-128 might be beneficial to mitigate the behavioral phenotype associated with these diseases.

## 1. Introduction

Noncoding DNA (ncDNA) represents a large portion of the human genome. Traditionally, it was believed that only a small portion of it had important biological functions; however, over the last 50 years, it has become apparent that even nonprotein coding regions of the genome are transcribed into noncoding RNA (ncRNA), which contribute to key functions associated with development and homeostasis. While it is likely that many of these transcripts are simply junk, it is conceivable that many more functional noncoding RNAs are yet to be discovered [1]. The discovery of specialized ncRNAs involved in gene silencing triggered a search that led to the identification of other ncRNAs and their associated role in biological functions; indeed, it is now established that ncRNAs play a key role in multiple biological functions, often in a tissue-dependent manner [2]. ncRNAs can regulate gene expression using multiple mechanisms, including directly binding transcription factors or RNA polymerase II (Pol II). In this case, ncRNAs can guide RNA Pol II to bind to the promoter of protein-coding genes, competing for enhancers [3]. Another mechanism by which ncRNAs regulate gene expression is by binding to a target mRNA thereby precluding its translation. For example, ncMyoD, an ncRNA activated by myogenic differentiation (MyoD), directly binds to its target mRNA (IGF2-mRNA-binding protein 2) reducing its translation and negatively regulating the downstream pathway activated by IMP2 [4].

MicroRNAs (miRNAs or miRs) use complementary mechanisms to regulate critical cellular processes. Scientists have agreed on the characteristics that must be met for a sequence to be considered miRNA; those include the detection of a 22-nucleotide RNA molecule, identification of that molecule in a pool of complementary DNA made from RNA with specified sizes, phylogenetic conservation of the molecule and the presence of a hairpin, which has important regulatory roles. Typically, both the gene locus and precursor miRNA (pre-miRNA) of a miRNA is referred to as “mir”, while the mature miRNA product is designated “miR”; when miRNAs are closely related in terms of their sequence, they are given additional suffixes in form of numbers and letters to distinguish them (e.g., mir-128). Their expression is tightly regulated and often in a developmental and tissue-specific manner. Consistent with these observations, miRNA’s dysregulation is associated with the pathogenesis of several human diseases [5]. miRNAs are small (21–23 nucleotides long), single-stranded ncRNAs that regulate protein production by inducing the degradation of target mRNAs or by inhibiting their translation [6]. By doing so, miRNAs actively regulate various cellular processes including cell differentiation, development, cellular aging, and neurodegeneration [7]. Specifically, they promote the cleavage of transcripts and repress translation mainly by base pairing within the 3′-untranslated regions (UTRs) of cognate mRNAs. An almost perfect match (i.e., minimal number of mismatches), between a miRNA and its target mRNA, leads to the degradation of the mRNA by activating the Argonaute complex [8]; in contrast, if the base pairing between the miRNA and the target mRNA is imperfect (i.e., a high number of mismatches), the miRNA inhibits translation initiation [9]. Converging evidence from multiple fields of studies has helped to understand the biogenesis and mechanisms of action of miRNAs supporting their role in human physiology [7]. Despite these observations, the exact biological functions of the majority of miRNAs known to date are less clear. One can easily predict that with technological advancements and increased interest in miRNA biology, the mechanisms of action of selective miRNAs will come to light in the near future [1]. Attainment of a greater knowledge of miRNA-mediated mechanisms of regulation of gene expression will enable their efficient therapeutic targeting in various human disorders, from cancer to neurodegeneration.

## 2. The Role of ncRNAs in Neurodegenerative Diseases

Growing evidence indicates that ncRNAs are involved in the pathogenesis of several neurodegenerative disorders (NDDs) [10]. Indeed, in recent years, ncRNAs have become key molecules implicated in almost every process leading to the onset and progression of NDDs. The association between ncRNAs and disease pathogenesis has also been suggested in Parkinson’s disease (PD) [11], Huntington’s disease (HD) [12], Alzheimer’s disease (AD) [13], and other neurodegenerative disorders [14]. Overall, six major ncRNA-mediated mechanisms are likely to contribute to the neurodegenerative process: epigenetic regulation, regulation of gene expression by RNA interference (RNAi), alternative splicing, mRNA stability, translation regulation, and molecular snare [15].

### 2.1. Epigenetic Regulation

Recently, it has become evident that ncRNAs play a major role in selective mechanisms associated with epigenetic modifications through which they regulate the expression of a single gene (e.g., by changing the epigenetic profile of its promoter or by regulating chromatin architecture) [16]. One way by which miRNAs can influence epigenetic mechanisms is by regulating the activity of chromatin remodeling enzymes, which in turn can influence the epigenetic landscape of large parts of the genome [17]. For example, miRNAs can induce chromatin remodeling through the regulation of histone modifications. According to one study, miR-206 specifically inhibits HDAC4, which in turn stimulates the release of fibroblast growth factor binding protein 1 and raises miR-206 expression levels to promote reinnervation at the neuromuscular junction [18]. Another example is represented by Argonaute-1 (AGO1), Polycomb group (PcG) component EZH2, and tri-methyl histone H3 lysine 27 (H3K27me3), which are directed to associate with the POLR3D promoter by miR-320, demonstrating the existence of an epigenetic mechanism of miRNA-directed transcriptional gene silencing in human cells [19]. Importantly, the regulatory effects of many ncRNAs still await systematic experimental validation in various neurodegenerative models [20].

### 2.2. RNA Interference (RNAi)

RNAi refers to a series of processes directed by miRNAs and small-interfering RNAs (siRNAs) that are aimed at reducing protein translation. Endogenous RNAi is a method that cells use to control mRNA expression. It is an evolutionarily conserved process in which double-stranded RNA sequences cause post-transcriptional gene repression, usually by destroying specific mRNA molecules. To this end, miRNAs and siRNAs can bind to selective mRNA targets thereby regulating their translation [21]. RNAi can be highly selective for a target mRNA, and as such is becoming a compelling molecular tool to regulate gene expression. To this end, reducing the expression of mutant genes associated with neurodegenerative disorders is an exciting and often underexplored approach. The RNase III endonuclease Dicer is a crucial molecular component shared between miRNA-mediated and siRNA-mediated mechanisms. RNAi-mediated gene silencing comprises a sequential biochemical process: Dicer turns dsRNAs to siRNAs, which are then integrated into the RNA-induced silencing complex (RISC) and guided to target mRNAs, thus effectively triggering their degradation. Dicer has a role in the generation of miRNAs, siRNAs, and the AGO protein family, a key member of the RISC [22]. The binding of miRNAs to their targets, which is mediated by RISC, leads to either mRNA decay or translational inhibition. RNAi-based therapeutic approaches have been developed for several neurodegenerative disorders. A crucial illustration is provided by HD and cerebellar ataxia, both of which are PolyQ diseases as they are linked to excessively long CAG repeats. siRNA-based techniques are aimed at targeting the gene with the extended CAG repeats; however, this strategy is limited since the siRNA may also have an impact on the wild-type gene, which may lead to unwanted side effects [23]. Another interesting example of RNAi-based therapy is that applicated to spinal muscular atrophy (SMA); particularly, the approach enables modification of defective transcripts from the survival motor neuron (SMN2) gene to induce gene transcription and translation into the SMN protein and restore its function. Notably, given the success of this approach in clinical trials, the FDA approved SPINRAZATM (Nusinersen) as the first and only genetic treatment to date for SMA [24].

### 2.3. Alternative Splicing

ncRNAs may affect pathology in neurodegenerative disorders by shifting the splicing profiles of transcripts. A classic example is represented by long non-coding RNA (lncRNA) 51A, which induces a splicing shift of the sortilin-related receptor 1 (SORL1). Physiologically SORL1 regulates APP trafficking through endocytic and secretory compartments. lncRNA51A-mediated decrease of the primary variant of SORL1 leads to altered APP trafficking and an increase in Aβ generation. Notably, 51A levels are higher in plasma and brains of AD patients [25].

### 2.4. Translational Regulation

As mentioned above, ncRNAs directly regulate the translation of target mRNA molecules. For example, BC200, a neuronal-specific ncRNA, is transported to dendrites, where it regulates the local translation of several mRNAs [26]. Considering that the local translation in dendrites is fundamental for synaptic plasticity and memory consolidation and that BC200 levels are markedly increased in the human AD brain, it is tempting to speculate that BC200 might be involved in the synaptic and memory deficits associated with AD [27].

### 2.5. Molecular Snare

The transcription of ncRNAs’ near gene promoters and enhancers makes it easier for transcription factors to bind to their DNA targets, pointing to potential roles for ncRNAs associated with promoters and enhancers in the recruitment of transcription factors to their binding sites [28]. To convey an RNA or a transcription factor to selective cellular compartments, ncRNAs might operate as molecular traps. The number of target molecules that a single ncRNA can sequester in a specific cellular setting, as well as the relative abundance of the ncRNA transcription and its targets, are crucial elements that govern the results of such interactions [29]. Generally, the soluble N-ethylmaleimide-sensitive factor attachment receptor proteins complex (SNARE) represents an efficient and controllable complex that drives the fusion of biological membranes [30] and facilitates exosomal miRNAs’ release. Other important proteins that are required for exocytosis, including Munc-18, Munc-13, and complexins, interact with SNAREs to enhance the fusion probability between vesicles and presynaptic membranes [30].

### 2.6. ncRNA in AD

Growing evidence suggests that miRNAs and lncRNAs (but even other ncRNAs) are involved in AD, the most common neurodegenerative disorder (see below). miRNA profiling studies indicate that miRNAs are abnormally expressed in AD patients and animal models. To this end, it appears that many of these abnormally expressed ncRNAs regulate Aβ, tau, inflammation, and cell death, among other pathways linked to the onset and progression of AD [31]. Compared with miRNAs, lncRNAs have a long strand and more subsequences. A classic example of lncRNAs is represented by BACE1 antisense transcript (BACE1-AS), transcribed from the antisense strand of BACE1; its levels are increased in AD [32] and regulate BACE1 mRNA and protein expression. BACE1-AS forms RNA duplexes with BACE1 mRNA, which results in higher stability of the mRNA and an increase in BACE1 levels and activity [33]. In contrast, downregulation of lncRNA BACE1-AS expression by siRNA reduces BACE1′s ability to cleave APP thereby lowering Aβ [34].

## 3. The Role of miR-128

### miR-128: Physiological Role and Origin

miR-128 is highly expressed in mammalian brains [35]; it is encoded with two independent genes, miR-128-1 and miR-128-2. In mice and humans, miR-128-1 is located on chromosomes 1 and 2, respectively. In contrast, miRNA-128-2 is located on mouse chromosome 9 and human chromosome 3. In mice, ~80% of the miR-128 transcript is from the miR-128-2 gene, while the remaining 20% is from the miR-128-1 gene [36]. miR-128 expression increases during development and peaks in adulthood, where its expression in the neocortex is ~4-fold higher than in the cerebellum [35,37].

miR-128 plays a key function in the development of the nervous system and the regulation of various neural functions, and it has also been associated with different types of cancer [38]. miR-128 expression is increased in the heart where it is involved in cardiomyocyte proliferation and heart regeneration [39]. miR-128 decreases the expression of pericentriolar material 1 (PCM1), which is involved in the proliferation and differentiation of neural progenitor cells (NPCs). To this end, overexpression of miR-128 facilitates NPC differentiation into neurons [40]. Overexpression of miR-128 suppressed the translation of PCM1, and the knockdown of endogenous PCM1 phenocopied the observed effects of miR-128 overexpression. Furthermore, concomitant overexpression of PCM1 and miR-128 in NPCs rescued the phenotype associated with miR-128 overexpression, enhancing neurogenesis but inhibiting proliferation [40]. Interestingly, miR-128 might act both as a tumor suppressor and an oncogene even though the exact mechanisms of action are still being investigated. The involvement of miR-128 in cell death has been demonstrated in neuroblastoma cells; particularly, overexpression of miR-128 causes morphological changes in SH-SY5Y neuroblastoma cells, as well as a significant increase in cell number. Consequently, transcriptome analysis of cells transfected with miR-128 highlighted changes in the expression of genes involved in cytoskeletal organization, apoptosis, cell survival, and proliferation [41].

In glioma, miR-128 levels are reduced, which hinders cell proliferation and self-renewal, while it facilitates apoptosis and suppressed motility. These effects appear to be mediated by the effects of miR-128 on Bmi-1. This is the first example of specific regulation by miRNAs of a neural stem cell self-renewal factor, implicating the role of miR-128 as important biological and therapeutic targets against the “stem-cell-like” characteristics of glioma [42]. Further related research may aid the development of new therapeutic strategies against glioma [43]. miR-128 is differentially expressed in various tumors, and its aberrant expression can be observed in many other malignancies, such as gastric carcinoma, lung cancer, pancreatic cancer, and hepatocellular carcinoma, contributing to tumorigenesis and metastasis. Some studies show that miR-128 acts as a tumor suppressor; indeed, up-regulation of miR-128 reduces neuroblastoma cell motility and invasiveness and could curb the proliferation and invasion of prostate cancer cells [44,45]. Similarly, miR-128 targets vascular endothelial growth factors, and by doing so, it inhibits tumorigenesis, angiogenesis, and lymphangiogenesis [46]. However, miR-128 may also act as an oncogene, as suggested by numerous other studies. For example, Mets and colleagues found that miR-128 is a candidate oncogenic miRNA in T-cell acute lymphoblastic leukemia, which targets the PHF6 tumor suppressor gene [47]. Along these lines, Zhuang and colleagues highlighted that the expression of miR-128 in hepatocellular carcinoma tissues was up-regulated compared with its expression in adjacent nontumor tissues [48]. miR-128 has also been connected to the progression of breast cancer, even though the specific pattern of expression and underlying molecular basis underlying its link to this disorder has not yet been fully explored [49].

## 4. miR-128 in Neurodegenerative Disease

### 4.1. miR-128 in Alzheimer’s Disease

AD is the most common form of dementia in the elderly. Clinically, AD manifests with a progressive cognitive decline, which in part arises from AD patients’ inability to form and retrieve semantic and episodic memories, spatial orientation deficits, and impediments in language and social interaction [50]. AD pathology is associated with the accumulation of Aβ plaques and tau tangle. These devastating events lead to the loss of synapses and alterations in synaptic plasticity, which occur in several brain areas including the hippocampus and entorhinal cortex, which are the first to be affected in AD [51,52]. Eventually, neurons in several brain regions degenerate [53,54,55]. miRNAs may contribute to AD pathogenesis by altering the production of proteins that regulate synaptic plasticity and neuronal transmission or affect cellular survival. Indeed, there is an inverse correlation between the expression of several miRNAs involved in synaptic plasticity and/or cell survival and the progression of AD neuropathology both in human brains and mouse models [56,57]. miR-128 is highly expressed in the brain where it contributes to neuronal plasticity [58]. Its steady-state levels are increased in the hippocampi of AD brains compared to healthy controls [59,60]. Interestingly, a recent report has indicated that miR-128 levels are also upregulated in sera of AD patients and that its changes positively and significantly correlate to the Mini-Mental State Examination scores [61]. While further studies are needed to confirm these results, sera levels of miR-128 could potentially become a valid diagnostic biomarker for AD, as observed in Figure 1 [62]. Data from mouse models further confirm a primary role for miR-128 in AD pathogenesis. To this end, knocking out miR-128 in 3xTg-AD mice, a widely used mouse model of AD [63], improved cognitive deficits and reduced Aβ pathology [58]. Similar results were obtained using a different mouse model of AD [64]. Despite these findings, the mechanism by which miR-128 is linked to AD remains unknown. It may contribute to AD because of its role in neuronal plasticity. To this end, miR-128 downregulation has been shown to increase cultured cortical network excitability [65] and regulate neuronal excitability and several genes related to neural plasticity [66]. Particularly, miR-128 knockdown causes a small but significant decrease in MAPK3/1 activation; the small effects on the MAPK3/1 pathway activation could be due to the low transduction efficacy obtained during the experiments. This would suggest that inhibition of miR-128 in mature culture also leads to neuronal overexcitability and is not ascribed to an effect of network development [36]. Overall, whether the upregulation of miR-128 in the brain and sera of AD is a consequence of a specific pathogenic event occurring during AD progression or it contributes to AD onset remains to be determined.

### 4.2. miR-128 in Parkinson’s Disease

PD is one of the most common neurodegenerative diseases among the elderly. One classical hallmark of PD is the gradual degeneration of dopaminergic neurons in substantia nigra pars compacta. This loss of dopaminergic neurons is associated with dopamine deficiency in the striatum, which in turn contributes to the classical motor phenotype of PD patients. A major research effort over the last several decades has led to considerable advances in the understanding of genetic and environmental factors, as well as the underlying molecular mechanisms linked to the onset and progression of the disease [67]. About 10% of PD cases are inherited and linked to genetic mutations in the SNCA (α-synuclein), PARK2 (Parkin), PINK1 (PTEN-induced kinase 1), PARK7 (DJ-1 deglycase protein), LRRK2 (repeated rich kinase leucine), and ATP13A2 (type ATPase 13A) genes. Sporadic PD is associated with environmental factors that contribute to mitochondrial dysfunction, such as pesticides and heavy metals. Therefore, PD is often modeled in vivo with MPTP/MPP + (1-methyl-4-phenyl-1,2,3,6-tetrahydropyridine) or 6-OHDA (6-hydroxydopamine). These compounds selectively cause mitochondrial dysfunction and oxidative stress in dopaminergic neurons [68].

Clinically, PD presents both motor symptoms including bradykinesia, postural disturbances, muscle stiffness and dystonia, and nonmotor symptoms including sleep disturbances, cognitive impairment, dementia, depression, and anxiety. Neuropathologically, PD is characterized by the presence of α-synuclein-positive aggregates and Lewy bodies in nigrostriatal DA neurons, which are often associated with autophagosome accumulation and reduction of lysosomal markers [69]. These data suggest a defect in lysosome-mediated clearance of the α-synuclein aggregates. Currently, only symptomatic treatments are available for PD, which are not neuroprotective and do not slow disease progression or reverse neurodegeneration. These therapies mainly include drugs such as levodopa, dopamine agonists, catechol-O-methyl transferase inhibitors, and monoamine oxidase B inhibitors. However, long-term use of these pharmaceuticals is linked to the development of serious side effects such as dyskinesia [70]. Therefore, there is great interest in understanding the pathogenesis of PD to develop novel and effective treatments.

Several miRNAs, including miR-128, have been reported to indirectly regulate α-synuclein levels by targeting the expression of other genes. Overexpression of miR-128 leads to the repression of autophagy and the inhibition of the transcription factor TFEB and its related targets thereby exacerbating the vulnerability of DA neurons to α-synuclein toxicity, which will eventually cause loss of DA neurons and the development of motor deficits (Figure 2) [71].

Moreover, the increased expression of miR-128 appears to be involved in the regulation excitability of DA neurons by suppressing the expression of various ion channels and signaling components of the extracellular network of signal-regulated kinase 2 (ERK2). The hyperactivation of ERK2 causes a concomitant increase in the sensitivity of the D1 neuron to dopamine, which is a major cause of dyskinesia, a side effect of L-Dopa treatment in PD. MiR-128 deficiency in striatal D1 neurons mimics the hypersensitivity of D1 neurons in mice with PD-like phenotype. Conversely, the effect of increasing miR-128 expression in adult neurons protects mice from abnormal motor activities associated with chemically induced PD and seizures [36]. Further strengthening the link between miR-128 and PD are studies showing that miR-128 negatively regulates the axin-1 inhibitor protein (AXIN1), an inhibitor of the Wnt/β-catenin signaling pathway involved in neuronal differentiation through the modulation of β-catenin levels. The overexpression of miR-128 and consequently the downregulation of AXIN1 would seem to have a protective effect, reducing apoptosis of DA neurons, as shown in Figure 2 [72]. Additional results demonstrated that activation of the HIF-1α/miR-128-3p pathway activates the Wnt/β-catenin signaling leading to an inhibition of hippocampal neurodegeneration in an MPTP mouse model. HIF-1α and miR-128-3p are downregulated in the hippocampus of mouse models of PD, while AXIN1 is highly expressed. HIF-1α is an important transcription factor whose increased expression may play a neuroprotective role in a PD model [73]. It is essential to fully understand the mechanisms of dysregulated miRNAs in disease pathogenesis as they and their targets may have great diagnostic and therapeutic potential for PD.

Despite the importance of miRNA research, the currently available literature on the role of miRNAs in PD is lacking, and the results of miRNA studies are often inconsistent. In addition, the current experimental approaches used in miRNA research have many limitations, including the lack of a standardized protocol to analyze miRNA expression profiles, which may slow down the progression of miRNA research.

### 4.3. miR-128 in Huntington’s Disease

HD is an inherited disease caused by the abnormal expansion of a repeated trinucleotide sequence (CAG) in the huntingtin gene (HTT), translated into a long polyglutamine (poly Q) tract, which causes misfolding and aggregation of the mutant huntingtin (mHTT). The accumulation of mHTT is thought to be responsible for neurodegeneration. The presence of the polyQ expansion acts as a toxic gain of function and it converts HTT from a neuroprotective protein to a neurotoxic one [74]. Notably, the length of the CAG repeats inversely correlates with the age of disease onset. The length of CAG repeats in asymptomatic people is between 6 and 35. Patients with repeat lengths between 37–40 may have no symptoms or late onset of the disease. On the other hand, repetitions above 60 are likely to have a very severe phenotype and juvenile onset [75]. Symptoms of HD include severe motor dysfunctions (chorea, bradykinesia, and dystonia) and cognitive impairments, which include changes in attention, executive functions, and memory. In addition, HD patients may manifest psychiatric, sleep, and circadian disorders; weight loss often associated with atrophy of skeletal muscles; and peripheral immune system alterations [76]. While there are no therapies to prevent the onset or slow the progression of HD, clearance of mHTT could represent a valid therapeutic strategy to reduce HD-related symptoms. To this end, several HTT reduction approaches have been considered including the use of antisense oligonucleotides (ASOs), RNAi, ribozymes, DNA enzymes, and genome modification approaches [77].

Most of the studies in the literature focus on the analysis of miRNA expression profiles as HD markers in postmortem human brains and mouse models. Using qPCR, Kocerha et al. (2014) found a significant downregulation of miR-128 in the brains of HD monkeys compared to control animals [78,79,80]. Other studies have investigated the correlation between changes in miRNA expression and CAG repeat length in several brain regions. These studies revealed a link between CAG length and miRNA expression in the brain, with 159 miRNAs selectively altered in the striatum, 102 in the cerebellum, 51 in the hippocampus, and 45 in the cortex. Among the candidate miRNAs considered, miR-128 has been linked with various pathways linked to neuronal development and survival. Particularly, miR128 levels in four brain regions, at three different ages, were positively correlated with the CAG length of the Huntingtin gene in a mouse model of AD. This study provides a broad resource for length-dependent CAG repeat changes in miRNA expression in vulnerable and disease-resistant brain regions in HD mice and offers new information for further investigations of miRNAs in HD pathogenesis and therapy [81]. The role of miR-128 as a potential therapeutic target for HD has not been studied even though it might represent an ideal candidate as it appears to be involved in the modulation of many genes implicated in HD such as HIP1, SP1, and HTT itself [78]. Furthermore, miR-128 plays a key role in motor activity and neuronal excitability as its downregulation appears to be associated with the onset of epileptic seizures [82]. This is highly germane to HD research given that patients often develop seizures. Indeed, miR-128 ablation in neurons expressing the dopamine receptor 1 (D1 neurons) leads to juvenile hyperactivity and seizures, consistent with other reports involving D1 neurons in HD symptoms [83].

The main challenge is the selective and specific delivery of miRNAs to the CNS. As of today, the most promising approach used to deliver miRNAs across the BBB appears to be the use of adeno-associated viruses (AAVs). A classic example is represented by AMT-130, an investigational gene therapy approach aimed at modifying the progression of HD. Results in animal models are encouraging as they show that AMT-130 reduces the production of huntingtin, which in turn ameliorates the HD-like phenotype developed by the mice [84].

## 5. Current Contradictions and Future Challenges

Noncoding RNAs play a role in almost every aspect of an organism’s physiology, from the regulation of developmental stages to learning and memory, from reproduction to aging. Thus, it should not be surprising that their dysfunction may contribute to the onset and progression of neurodegenerative disorders such as AD, PD, and HD. However, it is surprising that one often finds apparently contradictory results in the literature, as we described above. For example, in PD, Decressac and colleagues showed that miR-128 reduced the expression of TFEB and exacerbated the PD-like phenotype developed by an animal model of PD [71]. In contrast, Zhou and colleagues came to diametrically opposed conclusions. They reported that increasing miR-128 expression reduces apoptosis of DA neurons [72]. While it is hard to pinpoint the possible reasons behind these contradictions, they often derive from the implementation of different methodologies. Decreassac and colleagues used an adeno-associated virus to increase the expression of miR-128 in the midbrain of an animal model. Zhou and colleagues used dissociated dopaminergic neurons for their studies. Similar contradictions can be found in the HD literature. For example, in a comprehensive study, Kocerha and colleagues reported that miRNA levels were decreased in both a nonhuman primate model of HD and in HD patients [78]. In contrast, Langfelder and colleagues reported a positive correlation between the number of CAG repeats and miRNA levels in a mouse model of HD [83]. There are two major differences that might account for the apparent contradicting results of these studies: (i) they used different species, and (ii) the former used a candidate approach and directly measured miR-128 while the latter used an unbiased approach, and the levels of miR-128 (together with other miRNAs) were found to be altered. Overall, these two studies highlight how the effects of miR-128 (and all the other miRNAs for that matter) might be context-specific, and one needs to take it into account before making generalized conclusions. In addition, there is an urgent need for standardized protocols to analyze miRNA expression profiles, which may slow down the progression of miRNA research. To this end, it would be beneficial if the field would find and agree upon standard protocols for the preparation of the samples. A somewhat successful attempt at this has been performed in the autophagy field [85,86].

While the noncoding RNA world has seen a great expansion over the last decade, more needs to be done to better understand the molecular mechanisms underlying the dysfunction of a selective noncoding RNA in specific diseases. It is not clear how changes in miR-128 may contribute to the various proteinopathies, and several fundamental questions remain unanswered: Are these changes a consequence of the neuropathological phenotype (e.g., accumulation of Aβ or α-synuclein), or are they involved in the disease pathogenesis, and its dysregulation triggers the neuropathological phenotype? What leads to the tissue-specific alterations of miR-128? What are the targets of miR-128? For example, although miR-128 has been linked to HD little is known about the mechanisms linking the changes in its expression with the pathogenesis of HD. Similar examples could be highlighted for other neurodegenerative diseases. Unveiling the causes underlying the alteration in expression of a specific noncoding RNA in each disease may lead to a better understanding of the disease mechanisms and eventually to the identification of novel therapeutic targets.

## Figures and Tables

**Figure 1 ijms-24-06024-f001:**
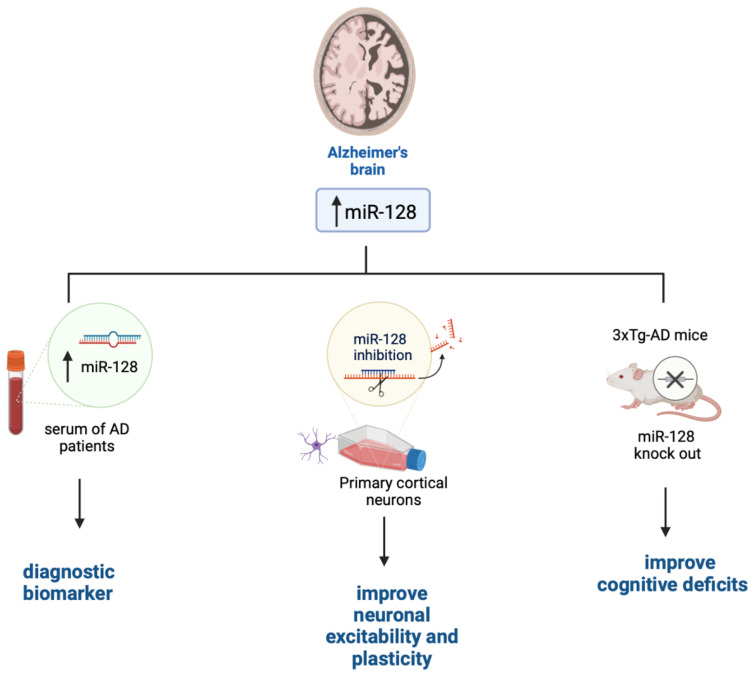
The role of miR-128 in AD. Figure created with BioRender.com accessed on 1 March 2023.

**Figure 2 ijms-24-06024-f002:**
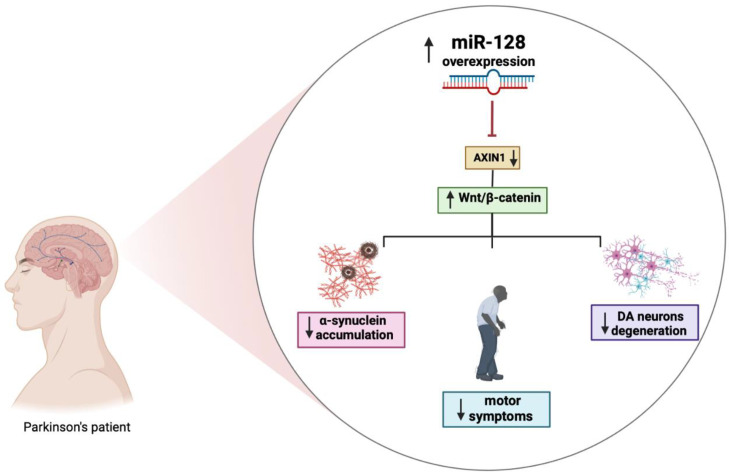
The role of miR-128 in PD. Figure created with BioRender.com accessed on 1 March 2023.

## Data Availability

Not applicable.

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
