# Peer review of "The Role of miR-128 in Neurodegenerative Diseases"

_ijms, 2023, doi:10.3390/ijms24076024_

Round 1

Reviewer 1 Report

In the review article „The Role of miR-128 in Alzheimer’s Disease and Related Disorders“, after the Introduction in which definition and types of non-coding RNAs are explained, the authors briefly described six major ncRNA-mediated mechanisms that probably contribute to the neurodegeneration: epigenetic regulation, regulation of gene expression by RNA interference (RNAi), alternative splicing, mRNA stability, translation regulation, and molecular snare, and gave examples on the role of several ncRNA in Alzheimer’s Disease (AD). Following presentation of miR-128 physiological role and origin, the authors gave short overview of known roles of miR-128 in Alzheimer’s disease, Parkinson’s disease and in Huntington’s disease, ending this review article with few remarks about future challenges regarding this field.

The article is fluent and easy to read and provides a short overview of the field for the reader who is not an expert in either the ncRNA world or non-degenerative diseases, and as such has certain value, although it lacks additional value to be classified as a good review article for the more demanding readers.

It is suggested that the authors modify the manuscript in such a way as to provide a more critical, comparative, and constructive evaluation of literature data and to identify gaps in existing studies for potential future research. To improve the article before it can be accepted for publication, it is suggested that the authors include an additional chapter in which they will give a detailed overview of the points they have mentioned as basic source of inconsistency among results, i.e., current experimental approaches used in miR-128 research.

Also, changes to the title, abstract and keywords are needed.

The title is not consistent with the abstract and keywords, nor with the content of the article itself. It is not clear why AD was put in the title, although it is no more represented in the article compared to Parkinson’s disease and in Huntington’s disease. The abstract is too general and does not give an accurate representation of the content of the article. Neither in the abstract, nor in the keywords, is miR-128 mentioned. Accordingly, the abstract does not even mention the role of miR-128.

Minor points:

1.       Figures are not mentioned in the text.

2.       The authors stated that Figure are created by Biorender. Did the authors design figures by themselves or was one of the available templates used? Do the authors have a license to publish figures made in Biorender and, given how the figures were made, is the used tool correctly cited?

3.       Authors should check whether they have inserted all the references in the appropriate places and correct it.

First example: Is ref 88.:

Kotowska-Zimmer, A.; Przybyl, L.; Pewinska, M.; Suszynska-Zajczyk, J.; Wronka, D.; Figiel, M.; Olejniczak, M. A CAG repeat targeting artificial miRNA lowers the mutant huntingtin level in the YAC128 model of Huntington's disease. Mol Ther Nucleic Acids 2022, 28, 702-715, doi:10.1016/j.omtn.2022.04.031.

really suitable for the statement:

„To this end, it would be beneficial if the field would find and agree upon standard protocols for the preparation of the samples. A somewhat successful attempt at this has been done in the autophagy field [87,88].

Another example is citing the article published in 2021 regarding facts that were known for some time i.e. :

statement:

„miRNAs are small (21-23 nucleotide long), single-stranded non–protein-coding RNAs that regulate protein production by inducing the degradation of target mRNAs or by inhibiting translation [6].“

Reference:

Chen, S.; Zhao, J.; Xu, C.; Sakharov, I.Y.; Zhao, S. Absolute Quantification of MicroRNAs in a Single Cell with Chemiluminescence Detection Based on Rolling Circle Amplification on a Microchip Platform. Anal Chem 2021, 93, 9218-9225, doi:10.1021/acs.analchem.1c01463.

Author Response

In the review article "The Role of miR-128 in Alzheimer’s Disease and Related Disorders“, after the Introduction in which definition and types of non-coding RNAs are explained, the authors briefly described six major ncRNA-mediated mechanisms that probably contribute to the neurodegeneration: epigenetic regulation, regulation of gene expression by RNA interference (RNAi), alternative splicing, mRNA stability, translation regulation, and molecular snare, and gave examples on the role of several ncRNA in Alzheimer’s Disease (AD). Following presentation of miR-128 physiological role and origin, the authors gave short overview of known roles of miR-128 in Alzheimer’s disease, Parkinson’s disease and in Huntington’s disease, ending this review article with few remarks about future challenges regarding this field.

The article is fluent and easy to read and provides a short overview of the field for the reader who is not an expert in either the ncRNA world or non-degenerative diseases, and as such has certain value, although it lacks additional value to be classified as a good review article for the more demanding readers.

It is suggested that the authors modify the manuscript in such a way as to provide a more critical, comparative, and constructive evaluation of literature data and to identify gaps in existing studies for potential future research. To improve the article before it can be accepted for publication, it is suggested that the authors include an additional chapter in which they will give a detailed overview of the points they have mentioned as basic source of inconsistency among results, i.e., current experimental approaches used in miR-128 research.

Also, changes to the title, abstract and keywords are needed.

The title is not consistent with the abstract and keywords, nor with the content of the article itself. It is not clear why AD was put in the title, although it is no more represented in the article compared to Parkinson’s disease and in Huntington’s disease. The abstract is too general and does not give an accurate representation of the content of the article. Neither in the abstract, nor in the keywords, is miR-128 mentioned. Accordingly, the abstract does not even mention the role of miR-128.

As suggested by reviewer, the authors rewrote the Abstract and changed the keywords and the title, to better elucidate the aim of the review and to give an accurate representation of the content of the article. Moreover, to discuss the inconsistencies found in the literature, the authors have expanded chapter 5.

Minor points:

- Figures are not mentioned in the text.

As suggested by reviewer, the authors mentioned the Figure in the text

- The authors stated that Figure are created by Biorender. Did the authors design figures by themselves or was one of the available templates used? Do the authors have a license to publish figures made in Biorender and, given how the figures were made, is the used tool correctly cited?

The authors designed the figures using Biorender by themselves, without recurring to available templates. The authors had a personal account, they are not part of a sitewide or departmental license. The authors better cited the tool used in the text.

      - Authors should check whether they have inserted all the references in the appropriate places and  correct

First example: Is ref 88.:

Kotowska-Zimmer, A.; Przybyl, L.; Pewinska, M.; Suszynska-Zajczyk, J.; Wronka, D.; Figiel, M.; Olejniczak, M. A CAG repeat targeting artificial miRNA lowers the mutant huntingtin level in the YAC128 model of Huntington's disease. Mol Ther Nucleic Acids 2022, 28, 702-715, doi:10.1016/j.omtn.2022.04.031.

really suitable for the statement:

„To this end, it would be beneficial if the field would find and agree upon standard protocols for the preparation of the samples. A somewhat successful attempt at this has been done in the autophagy field [87,88].

As suggested by reviewer, the authors checked the manuscript and put the appropriate references.

Another example is citing the article published in 2021 regarding facts that were known for some time i.e. :

statement:

„miRNAs are small (21-23 nucleotide long), single-stranded non–protein-coding RNAs that regulate protein production by inducing the degradation of target mRNAs or by inhibiting translation [6].“

Reference:

Chen, S.; Zhao, J.; Xu, C.; Sakharov, I.Y.; Zhao, S. Absolute Quantification of MicroRNAs in a Single Cell with Chemiluminescence Detection Based on Rolling Circle Amplification on a Microchip Platform. Anal Chem 2021, 93, 9218-9225, doi:10.1021/acs.analchem.1c01463.

As suggested by reviewer, the authors checked the manuscript and put the appropriate references.

Reviewer 2 Report

First of all, I would like to praise the authors for a quality review.

Non-coding abbrev. nc needs to be introduced in both DNA and RNA – see the introduction

Introduce other abbreviations throughout the article as well (lnc, …)

Also, make an introduction to why miRNAs are called miR (suggestion in line 47:  MicroRNAs (miRNAs or miRs)

Line 51 it doesn’t need to be called non-protein-coding, non-coding is enough

In the introduction, it would be beneficial to include a short section/commentary on the interactions of miRNA/ncRNA/lncRNA with neuropeptides in AD, PD, HD - https://doi.org/10.3390/bs12080262

Section 2.6 – revise for clarity its BACE overused

Section 3

-          Try to find a few citations for various types of cancer associated with miRs within the publishers’ portfolio lung, breast, prostate, don’t forget bladder, etc.

Section 4

-          Make figure 1 larger it loses readability when only in 100% size

Line 384 and 386 ncRNA – use abbreviation after first introduction

Author Response

First of all, I would like to praise the authors for a quality review.

The authors thank the reviewer for their appreciation of the review.

Non-coding abbrev. nc needs to be introduced in both DNA and RNA – see the introduction

Introduce other abbreviations throughout the article as well (lnc, …)

As suggested by reviewer, the authors introduced the abbreviations to make the text more understandable.

Also, make an introduction to why miRNAs are called miR (suggestion in line 47:  MicroRNAs (miRNAs or miRs)

As suggested by reviewer, the authors add an introduction to better clarify the nomenclature of miRNAs, in Introduction section.

Line 51 it doesn’t need to be called non-protein-coding, non-coding is enough

As suggested by reviewer, the authors corrected the sentence.

 In the introduction, it would be beneficial to include a short section/commentary on the interactions of miRNA/ncRNA/lncRNA with neuropeptides in AD, PD, HD - https://doi.org/10.3390/bs12080262

Section 2.6 – revise for clarity its BACE overused

As suggested by reviewer, the authors revised the overused BACE.

Section 3

      Try to find a few citations for various types of cancer associated with miRs within the publishers’ portfolio lung, breast, prostate, don’t forget bladder, etc.

The correlation between miR-128 expression and various types of cancer is not within the scope of the review, therefore the authors did not elaborate on this part, as suggested by the reviewers.

Section 4

    - Make figure 1 larger it loses readability when only in 100% size

      As suggested by reviewer, the authors increased the size of Figure 1 to improve the resolution.

    - Line 384 and 386 ncRNA – use abbreviation after first introduction

As suggested by reviewer, the authors used the abbreviation after first introduction.

Round 2

Reviewer 1 Report

I suggest accepting the revised version of the manuscript for publication in IJMS.

Author Response

I suggest accepting the revised version of the manuscript for publication in IJMS.

The reviewer did not have additional comments.

Reviewer 2 Report

Value of the article has substantially increased with resent changes. However, there are still a few issues:

I see the authors omitted my comment on a short section of AD,HD,PD connection to neuropeptides through miR-128.

Line 57 – single-stranded ncRNAs

A few spelling errors need to be corrected, singulars/plurals role-roles … RNA-RNAs, etc.

Author Response

We thank the reviewer for his input and for highlighting that the "Value of the article has substantially increased"

The reviewer raised some minor additional issues, which we have addressed. The answers to the reviewer's comments are in red below. The edits to this version of the manuscript were made using the Word Tracking System. 

I see the authors omitted my comment on a short section of AD,HD,PD connection to neuropeptides through miR-128.

We apologize for not clarifying this issue during the first cycle of reviews. We have done an extensive literature search from which it is evident that there is no connection between miR-128, neuropeptides and neurodegenerative diseases. Thus, we cannot write a small section on this topic. 

Line 57 – single-stranded ncRNAs. We apologize for the oversight. We have now fixed the issue.

A few spelling errors need to be corrected, singulars/plurals role-roles … RNA-RNAs, etc. We have carefully edited the entire manuscript to fix these issues.